# Chemical Composition of Essential Oil of *Cymbopogon schoenanthus* (L.) Spreng from Burkina Faso, and Effects against Prostate and Cervical Cancer Cell Lines

**DOI:** 10.3390/molecules28114561

**Published:** 2023-06-05

**Authors:** Bagora Bayala, Laetizia Liz Coulibaly, Florencia Djigma, Julio Bunay, Albert Yonli, Lassina Traore, Silvère Baron, Gilles Figueredo, Jacques Simpore, Jean-Marc A. Lobaccaro

**Affiliations:** 1Laboratoire de Biologie Moléculaire et de Génétique (LABIOGENE), Département de Biochimie-Microbiologie, Université Joseph KI-ZERBO, Ouagadougou 03 BP 7021, Burkina Faso; couleliz@gmail.com (L.L.C.); florencia.djigma@gmail.com (F.D.); ttl.lass@yahoo.fr (L.T.); simpore93@gmail.com (J.S.); 2Centre de Recherche Biomoléculaire Pietro Annigoni (CERBA), Ouagadougou 01 BP 216, Burkina Faso; yonlitheo@yahoo.fr; 3Institut Génétique, Reproduction & Développement, UMR CNRS 6293, INSERM U1103, Université Clermont Auvergne, et Centre de Recherche en Nutrition Humaine Auvergne, 28, Place Henri Dunant, BP38, F63001 Clermont-Ferrand, France; 4Ecole Normale Supérieure, Koudougou BP 376, Burkina Faso; julio.bunay@gmail.com (J.B.); silvere.baron@uca.fr (S.B.); 5LEXVA Analytique, Biopole Clermont-Limagne, F63360 Saint-Beauzire, France; gillesfigueredo@yahoo.fr

**Keywords:** *Cymbopogon schoenanthus*, essential oil, antioxidant, cytotoxic, antimigratory, cell cycle, cancer

## Abstract

The aim of this research was to evaluate the essential oil of *Cymbopogon schoenanthus* (L.) Spreng. (*C. schoenanthus)* from Burkina Faso in terms of cytotoxic activity against LNCaP cells, derived from prostate cancer, and HeLa cells, derived from cervical cancer. Antioxidant activities were evaluated in vitro. Essential oil (EO) was extracted by hydrodistillation and analyzed by GC/FID and GC/MS. Thirty-seven compounds were identified, the major compounds being piperitone (49.9%), δ-2-carene (24.02%), elemol (5.79%) and limonene (4.31%). EO exhibited a poor antioxidant activity, as shown by the inhibition of DPPH radicals (IC_50_ = 1730 ± 80 µg/mL) and ABTS^+.^ (IC_50_ = 2890 ± 26.9 µg/mL). Conversely, EO decreased the proliferation of LNCaP and HeLa cells with respective IC50 values of 135.53 ± 5.27 µg/mL and 146.17 ± 11 µg/mL. EO also prevented LNCaP cell migration and led to the arrest of their cell cycle in the G2/M phase. Altogether, this work points out for the first time that EO of *C. schoenanthus* from Burkina Faso could be an effective natural anticancer agent.

## 1. Introduction

Medicinal plants have been widely handled in traditional medicine for several centuries for the treatment of many health-related ailments [1]. Hence, in low-income countries such as Burkina Faso, medicinal plants are used with significant results to treat many pathologies, e.g., tumors, boils, or chronic wounds, especially in villages where modern health care is totally inaccessible. *Cymbopogon schoenanthus* (L.) Spreng (*C. schoenanthus*) is a herbal plant from Southern Asia and Northen Africa with fragrant foliage, suggesting the leaves are enriched in essential oils (EO). This herbaceous plant is well known and used in Burkina Faso in traditional medicine, as well as in other African and Asian countries. In fact, *Cymbopogon schoenanthus* is traditionally used for the treatment of various diseases. The rhizomes are used both internally, as a tonic, febrifuge, intestinal disinfectant, as well as externally, as a funeral disinfectant, anti-malarial and against guinea worm [1,2]. In Egyptian traditional medicine, this plant has a reputation as an antispasmodic and a diuretic. The plant is used in Sudan to treat gout, prostate inflammation, kidney disease and stomach pain [3]. In Burkina Faso, the plant is used in traditional pharmacopeia to treat skin ailments as well as coughs in infants and children [2]. Chemical characterization and in vitro analyses of EO of *C. schoenanthus* from various geographic areas have pointed out specific activities. EO extracted from the Sudanese plant has shown a high antiproliferative activity against human breast carcinoma and human colon adenocarcinoma cell lines [4]. EO extracted from the Saudi Arabian plant has strong protective effects against *Escherichia coli*, *Staphylococcus aureus*, methicillin-susceptible *S. aureus* and *Klebsiella pneumoniae* [1]. The authors identified eight main components, such as piperitone (14.6%), cyclohexanemethanol (11.6%), β-elemene (11.6%), α-eudesmol (11.5%), elemol (10.8%), β-eudesmol (8.5%), 2-naphthalenemethanol (7.1%) and γ-eudesmol (4.2%) [1]. EO of *C. schoenanthus* from Brazil presents an efficient anthelmintic activity, since the development of trichostrongylids obtained from naturally infected sheep is blocked in vitro [5]. Extracts from the Algerian plants show strong antioxidant activities [6,7]. So far, only one study has been performed on EO of *C. schoenanthus* from Burkina Faso [8]. Sawadogo et al. focused their work on the putative antifungal activities. The authors studied the inhibition of both mycelial growth of *Aspergillus parasiticus* and *Aspergillus flavus*, and activity of aflatoxins B2 and G1.

Cancer is a generic term referring to a large group of diseases which can affect any part of the body [9]. In Burkina Faso in 2020, according to GLOBOCAN, among the different types of cancer, prostate cancer and cervical cancer were among the most frequently diagnosed cancers, with 1132 new cases (14.6%) for cervical cancer and 997 new cases (23.2%) for prostate cancer [10].

Because EO in leaves of *C. schoenanthus* from Burkina Faso has been poorly studied so far, and the plants are used in Burkina Faso alone or in association in several recipes for the treatment of inflammatory, bacterial and tumoral diseases by traditional medicine, we aimed to evaluate the chemical composition of this EO to investigate its biological activities on the survival characteristics of LNCaP and HeLa cells, respectively derived from prostate and cervical cancers. Effects on the cell cycle and migration were also investigated in LNCaP cells.

## 2. Results and Discussion

### 2.1. Chemical Composition of the Essential Oil

Gas chromatography coupled with a flame ionization detector and mass spectrometry (Figure 1) identified 37 compounds within the EO of *C. schoenanthus*, for a percentage of 98.46% (Table 1).

The main compounds were piperitone (49.9%), *δ*-2-carene (24.02%), elemol (5.79%) and limonene (4.31%), representing almost 84% of the total composition. Monoterpene ketones, such as piperitone, represent up to 50% of the compounds. The levels of piperitone, a compound generally used for the production of synthetic menthol and thymol, is in accordance with the other studies with 59.1% [4] and 59.8% [8]. Interestingly, other authors have qualitatively and quantitatively described different compositions for EO extracted from *C. schoenanthus*. Hashim et al. [1] described that piperitone, cyclohexanemethanol, β-elemene, α-eudesmol and elemol were equally distributed in the EO. On the other hand, Katiki et al. showed that the major compound quantified by gas chromatography was geraniol (62.5%) [5]. We have hypothesized that these differences were due to the collection area and/or season.

### 2.2. Antoxidant Potential of the Essential Oil

The need for antioxidant compounds is important as oxidation is a process strongly involved in many pathologies. The inhibitory activities of DPPH and ABTS^+.^ radicals proportionally reflect the antioxidant activity. Since EO of *C. schoenanthus* was enriched in putative antioxidant molecules, inhibition of the DPPH and ABTS^+.^ radicals were measured. Concentrations necessary to reach 50% of the inhibition (IC_50_) were calculated as 1730 ± 80 μg/mL and 2890 ± 26.9 µg/mL for DPPH and ABTS^+.^, respectively (Table 2).

Compared to Trolox, which was used as a positive control, EO of *C. schoenanthu* was 900- and 300-fold less efficient on DPPH and ABTS^+.^ radicals, respectively, which led us to conclude that this EO has a really poor antioxidant activity. Interestingly, EO of *C. schoenanthus* from Sudan also has a weak antioxidant activity [4], conversely to the extracts collected from Algeria [11] and South Tunisia [12]. The main differences between our EO and those described were the amount of cis- and trans-pmeth-2-en-1-ols (0.81 and 0.55% for our study, vs. 23 and 14%, respectively) in the Algerian EO [11] and the amount of limonene (above 11% vs. 4% for the Burkina Faso) and α-terpineol (above 7% vs. 1% for the Burkina Faso) for the Tunisian EO [12].

### 2.3. Cytotoxic Activity of the Essential Oil

The imperfections in drugs currently used in therapy and the increasing problem of drug resistance have forced a search for new substances with therapeutic potential. Throughout history, numerous organisms have been rich sources of biologically active compounds [13]. To investigate whether *C. schoenanthus* could be such a source, the effects of EO of *C. schoenanthus* were evaluated on LNCaP (derived from metastatic prostate cancer) and HeLa (derived from cervical cancer) cell viability.

As shown in Figure 2, EO from *C. schoenanthus* decreased cell survival with relatively low IC_50_s of 135.53 ± 5.27 μg/mL and 146.17 ± 11.83 μg/mL on LNCaP and HeLa cells, respectively (Table 3), compared to the positive control, cisplatin (3.2 ± 0.5 and 5.2 ± 0.8 μg/mL on the cells, respectively, *p* < 0.001).

Mohamed Abdoul-Latif et al. have shown that EO of *C. schoenanthus* from Djibouti has more significant cytotoxic effects on A2780, A549, HCT116, HEK-293, JIMT-T1, K56s2, MIA-Paca2, MRC5, NCI-N87, PC3, RT4, U2OS and U87-MG cancer cell lines [14]. It should be noted that piperitone was totally absent in their EO, while 3-isopropenyl-5-methyl-1-cyclohexene and D-limonene were the main compounds. Conversely, Hakkim et al. pointed out that piperitone, present at 39% in EO from Oman, has strong cytotoxic activity in triple negative breast cancer and cervical cancer cell lines [15]. Piperitone representing almost 50% of the EO could be responsible for the results we obtained on LNCaP and HeLa cells. We have also associated the cytotoxic activity to the high amount of terpinenes, known to have anticancer properties [16]. Monoterpene ketones (50.1%) and monoterpene hydrocarbons (29.73%) could also explain this activity. However, compared to cisplatin, used as the control (Table 3), EO of *C. schoenanthus* is less efficient on LNCaP (135 vs. 3 µg/mL) and HeLa (146 vs. 5 µg/mL) cells. Altogether, it should thus be interesting to test the cytotoxic activity of piperitone, the major compound identified in our study, alone or in combination with the other compounds.

### 2.4. Effects of Essential Oil on Migration and Cycle of LNCaP Cells

Killing cells and blocking their migration are important pharmacological targets in the clinical management of tumors [17,18]. As cell migration is a hallmark in tumor development, we performed scratch cell assays to measure the effects of the tested EO on the migration of the LNCaP cells (Figure 3). While the wounded area was decreased by 32% (*p* < 0.001) after 72 h, EO used at the IC_50_ blocked cell migration. In fact, plants are known to have anti-migratory effects on cancer cells, as reported by Al-Maharik et al. [19] and Boonyanugomol et al. [20]. Effects that could be explained by the nature of the chemical compounds contained in the extracts of these plants.

Cancer cells can also be characterized by a strong activity of division [21]. We also investigated whether EO from *C. schoenanthus* could modify the cell cycle of LNCaP cells. Our experiments revealed a significant activity of EO on the distribution of the various phases of the cell cycle (Figure 4). An increased accumulation of cells in the subG1 phase was observed together with a 2-fold increase in the G0/G1 (from 35% to 70%, for 0 and 285 μg/mL of EO, respectively) and S (from 2% to 13%, for 0 and 285 μg/mL of EO, respectively) phases. In parallel, a significant decrease in G2/M phase was shown in cells treated with 285 μg/mL of EO compared to the control cells (from 63% to 17%, *p* < 0.01). It could thus be suggested that EO of *C. Schoenanthus* reduces cell proliferation. Whether an early apoptosis is induced needs to be demonstrated.

## 3. Materials and Methods

### 3.1. Plant Material and Essential Oil (EO) Extraction

Leaves of *C. schoenanthus* were collected at the National Institute of Applied Sciences and Technologies (IRSAT) in Ouagadougou, Burkina Faso (GPS location: 12°25′29.5″ N 1°29′14.3″ W). Identification and authentication were performed by Dr. A. Sereme (IRSAT/CNRST, Ouagadougou, Burkina Faso). A specimen was deposited in the herbarium of the Laboratory of Biology and Plant Ecology of University Joseph KI-ZERBO, publicly accessible under ID: 15936 and sample number 03 of September 2019. Fresh leaves of *C. schoenanthus* (1 Kg) were submitted to hydrodistillation using an alembic/Clevenger-type apparatus for 3 h as described previously [22]. EO was stored in airtight containers in a refrigerator at 4 °C until GC-FID and GC/MS analyses and biological tests. EO was therefore diluted in hexane (1/30, *v*/*v*) for GC/FID analysis.

### 3.2. Chemical Composition

Gas chromatography–flame ionization detector (GC/FID) analysis—Composition of EO was performed as previously mentioned [23]. Briefly, gas chromatography of hexane-diluted EO was performed on an Agilent gas chromatograph, model 6890 (Agilent, Palo Alto, CA, USA), equipped with a DB5 MS column (30 m × 0.25 mm, 0.25 µm film thickness). Hydrogene was used as carrier gas (1.0 mL/min flow rate). The oven temperature program was 50 °C (5 min) to 300 °C with an increasing temperature of 5 °C/min. The sample (1 mL) was injected in split mode (1:60), with injector and detector temperatures of 280 °C and 300 °C, respectively [22].

Gas chromatography–mass spectrometry (GC/MS) analysis—Mass spectrometry analyses of EO were performed on an Agilent gas chromatograph, model 7890, coupled to an Agilent MS, model 5975, equipped with a DB5 MS column (20 m × 0.20 mm, 0.20 mm film thickness). The oven temperature program was 50 °C (5 min) to 300 °C at 8 °C/min, with a 5 min hold, as described in [23]. Briefly, helium was used as the carrier gas (average flow of 1.0 mL/min). The oven temperature program was from 50 °C (3.2 min) to 300 °C at 8 °C/min, with a 5 min post run at 300 °C. Sample (1 µL) was injected in split mode (1:150), with injector and detector temperatures of 250 °C and 280 °C, respectively [22]. The MS worked in electron impact mode at 70 eV; the electron multiplier was set at 1500 V; the ion source temperature was 230 °C. Mass spectra data were acquired in scan mode in a *m*/*z* range of 33–450 [22].

Identification of components—The main compounds in the EO of *C. schoenanthus* were identified by comparison of their retention indices with those of the literature, determined in relation to a homologous series of n-alkanes (C8–C32) under the same operating conditions [23]. Instead of standard compounds, we performed retention indices and comparisons with the NIST library [24] or literature [25]. Component relative percentages were calculated based on GC peak areas without using correction factors [22,26]. The major identified compounds are indicated on Figure 1.

### 3.3. Cell Culture

LNCaP cells, derived from prostate cancer (ATCC # CRL-1740), or Hela cells, derived from cervical cancer (ATCC # CCL-2), were used. These cell lines were available at the GReD Institute (Université Clermont Auvergne, France) [27]. Cells were cultured and maintained at 37 °C in a chamber moistened with 5% CO_2_ in 75 cm^2^ flasks in RPMI-1640 or DMEM medium (Invitrogen, Oslo, Norway) supplemented with 10% fetal calf serum (FCS, Biowest, Nuaillé, France), 1% penicillin and 1% streptomycin (Invitrogen).

### 3.4. Antioxidant Activity

DPPH radical scavenging assay—As was already described [23], each 100 μL of serial dilutions of EO of *C. schoenanthus* starting at 3.8 mg/mL was mixed with 100 μL of DPPH (30 mg/L in methanol) and incubated for 30 min in darkness at room temperature. The absorbance was read at 517 nm against a blank (mixture without EO). Trolox (Sigma-Aldrich, L’Ile d’Abeau, France) was used as a positive control. The radical scavenging activity was calculated according to the formula: (absorbance blank—absorbance sample)/absorbance blank) × 100. The concentration of extract with the ability to scavenge 50% of the DPPH radicals was then determined graphically and expressed as μg of EO/μg of DPPH [27].

ABTS^+.^ radical cation decolorization assay—The spectrophotometric analysis of ABTS^+.^ scavenging activity was performed using 96-well plates with 50 μL of ethanolic solution of EO of *C. schoenanthus* at an initial concentration of 3.8 mg/mL, added to 200μL of freshly prepared ABTS^+.^ solution, as previously described [23,27]. Briefly, ABTS^+.^ solution was prepared by dissolving 10 mg of ABTS in 2.6 mL of distilled water in which 1.7212 mg of potassium persulfate was added. ABTS was then incubated at room temperature for 12 h and diluted with ethanol in order to obtain an absorbance of 0.70 ± 0.02 to 734 nm. Trolox (Trolox, Sigma-Aldrich) was used as a positive control. The 96-well plates were then incubated in the dark at room temperature for 15 min and the absorbance read at 734 nm. The activity of EO of *C. schoenanthus* on the radical cation ABTS^+.^ was expressed in micromoles Trolox equivalent per gram of EO (µmol TE g^−1^) [27].

### 3.5. Measurement of Cell Survival

The 3[4,5-dimethylthiazol-2-yl]-diphenyltetrazolium bromide (MTT) assay (Sigma-Aldrich) was used to measure the mitochondrial activity, which reflects the number of viable cells [22,23]. Briefly, 12,500 cells were seeded for 24 h in 96-well plates. EO of *C. schoenanthus* was added at various concentrations for 72 h. The number of living cells is directly proportional to the intensity of the violet color measured quantitatively by spectrophotometry using a Thermo Fisher Scientific SN 1510-02948 microplate reader spectrophotometer at 570 nm. Three independent experiments were performed in octuplicate for each cell line.

### 3.6. Anti-Migratory Activity

The effects of EO of *C. schoenanthus* on the migration of LNCaP and HeLa cells were evaluated with the in vitro wound healing assay. Cells were seeded in 6-well plates at a rate of 35 × 10^4^ cells per well in a final volume of 2 mL of complete medium for 24 h. A scratch was performed on the cell monolayer with a sterile micropipette tip. Detached cells were then removed with 1X phosphate buffered saline (Life Technologies, Carlsbad, CA, USA). Cells were next grown in medium containing EO of *C. schoenanthus* at the concentration corresponding to the IC_50_. Multiple images were then taken for 72 h with a Zeiss AxiObserver 7 inverted microscope equipped with a Colibri 7 LED source, which made it possible to determine the effect of the EO on LNCaP cell migration.

### 3.7. Flow Cytometry Analysis

LNCaP cells (3 × 10^5^) were seeded in 6-well dishes and treated with EO after 24 h at doses of 0 µg/mL (control; DMSO, 0.001), 143 µg/mL and 285 µg/mL, respectively, for 72 h at 37 °C. After the treatment, cells were harvested with trypsine, centrifuged and fixed with paraformaldehyde (4%) for 15 min at room temperature and then washed with PBS. 10^6^ cells were prepared in suspension, centrifuged and the supernatant removed. Then, 0.2 mL of FxCycle^TM^ PI/RNase staining solution (Invitrogen, Oslo, Norway) was added to each tube and mixed. The samples were incubated for 30 min at room temperature, protected from light, and then analyzed by FACS using excitation at 488 nm, and the emissions were collected using a 585/42 bandpass filter.

### 3.8. Statistical Analysis

The data were analyzed by analysis of variance followed by the Tukey multiple comparison test. The analyses were performed using XLSTAT 7.1 software. *p* < 0.05 was used as a criterion for statistical significance.

## 4. Conclusions

*C. schoenanthus* is a medicinal plant used in Burkina Faso alone or in association in several recipes for the treatment of inflammatory, bacterial and diseases described as tumoral by traditional medicine. The interest of using EO is that it is a mixture of various natural compounds, and thus it could target several signaling pathways. While low doses of a single compound may not achieve the full anti-cancer action, a mixture of compounds, as we have tested in cell culture, may specifically synergize in the prevention/therapy of cancer with negligible side effects. Altogether, this is what our work has tried to demonstrate regarding the effects of EO of *C. schoenanthus* from Burkina Faso on LNCaP and HeLa cells. The chemical composition points out the richness of EO in piperitone and terpenes. While the antioxidant capacity of the EO is quite low, cytotoxic activities on LNCaP and HeLa cells are significant. Besides, EO also prevents migration of LNCaP cells and leads to the arrest of their cell cycle in the G2/M phase. Altogether, this work constitutes a scientific basis to start investigating the potential use of *C. schoenanthus* from Burkina Faso in the management of tumors. Clearly further studies will be necessary to evaluate the effects of the main compounds and to search for possible synergistic, additive or antagonistic effects. This will also be necessary to determine the possible mechanism of the cytotoxic activity of piperitone, its main compound. Besides, and as suggested for the use of *Rhus coriaria* L. (Sumac) for the treatment of breast cancer [28], further investigations are needed to confirm that EO of *C. schoenanthus* from Burkina Faso can be used for the traditional management of tumors, such as the identification of mechanisms involved in the cytotoxic activity and in vivo studies on pre-clinical models. These studies will allow the safety profile to be defined and to putative adverse side effects of the EO to be identified.

## Figures and Tables

**Figure 1 molecules-28-04561-f001:**
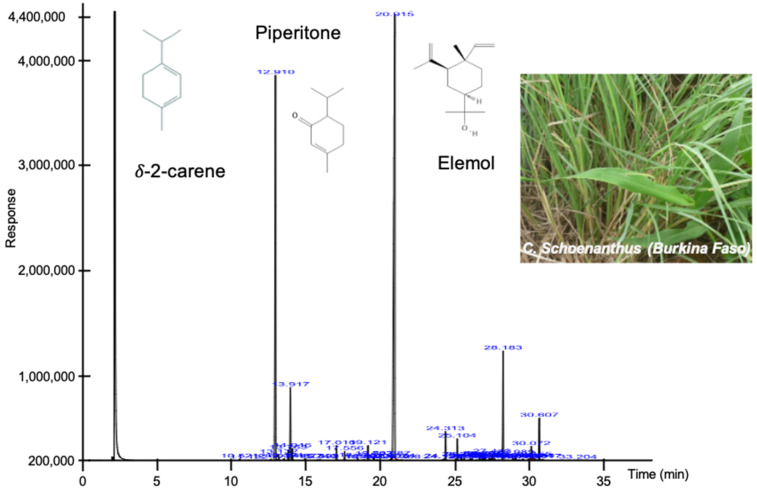
Chromatogram of identified compounds of *C. schoenanthus* EO. Structures of the three major identified compounds are drawn. Essential oil was extracted by hydrodistillation using an alembic/Clevenger-type apparatus for 3 h and stored in the bottle at 4 °C. Inset, photograph of *Cymbopogon schoenanthus* taken by Dr. Bagora BAYALA in Ouagadougou, Burkina Faso. GPS location: 12°25′29.5″ N and 1°29′14.3″ W.

**Figure 2 molecules-28-04561-f002:**
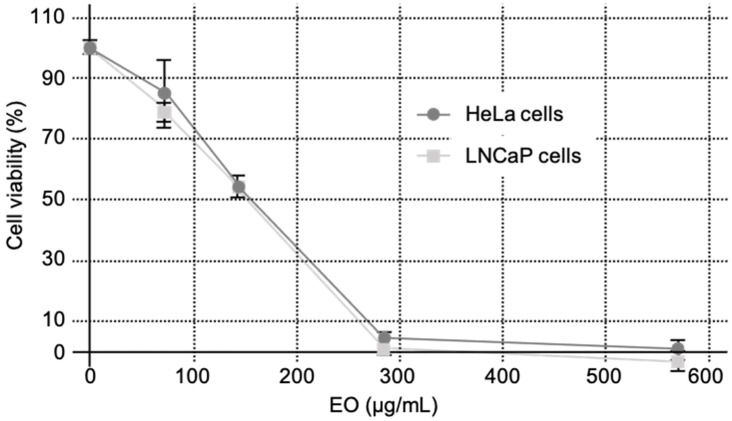
Effects of *C. schoenanthus* EO on LNCaP and HeLa cell survivals. LNCaP and HeLa cells were treated 24 h after seeding with various concentration of EO for 72 h. Values are expressed as mean values ± SD. *n* = 3 independent experiments in sextuplicate.

**Figure 3 molecules-28-04561-f003:**
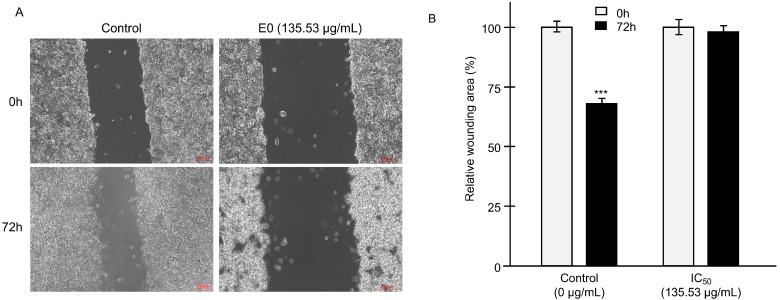
Activity of *C. schoenanthus* EO on cell migration. (**A**) Significant picture of scratch cell assay. (**B**) Relative wound healing area remaining after the incubation with EO at the IC_50_. Values are expressed as mean values ± standard deviation; *n* = 3 independent experiments; ***, *p* < 0.001 values significantly different compared to IC_50_ of EO after 72 h of induction 0 h.

**Figure 4 molecules-28-04561-f004:**
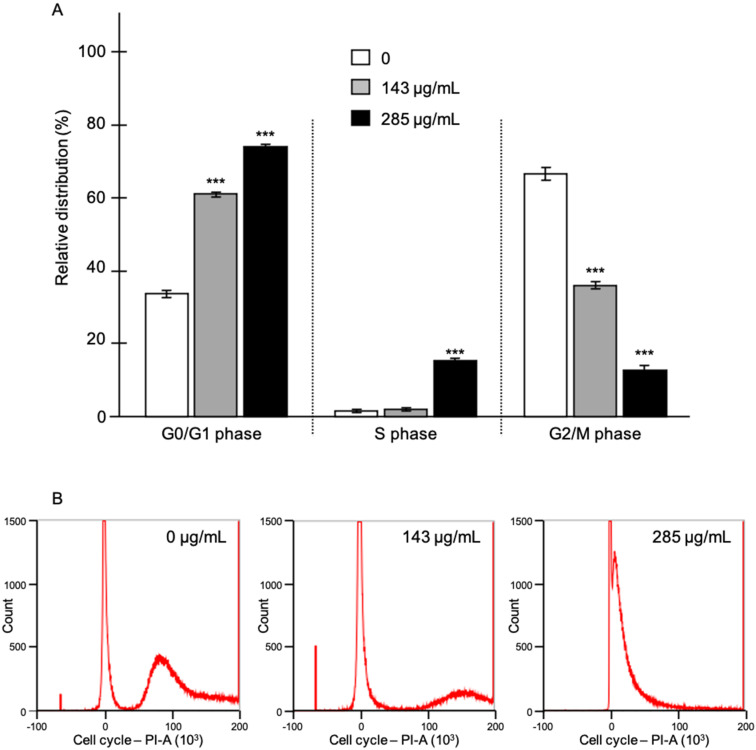
Effect of *C. schoenanthus* EO on LNCaP cell cycle. (**A**) LNCaP cells were incubated with control (DMSO 1/1000) and without EO (0 µg/mL) or with EO (142 or 285 μg/mL) diluted in DMSO for 72 h. Values are expressed as mean values ± standard deviation; *n* = 3 independent experiments in triplicate; ***, *p* < 0.001 values significantly different compared to vehicle-treated condition; EO, essential oil. Flow cytometry plots are shown in (**B**).

**Table 1 molecules-28-04561-t001:** Chemical composition of essential oil of *Cymbopogon schoenanthus*.

Chemical Compounds	Retention Time (min)	Percentage (%)
α-pinene ^Mh^	10.521	0.11
1,8-Cineol dehydro ^Me^	12.603	0.11
δ-2 carene ^Mh^	12.910	24.02
α-phellandrene ^Mh^	13.132	0.38
α-terpinene ^Mh^	13.494	0.13
P-cymene ^Mh^	13.763	0.58
Limonene ^Mh^	13.917	4.31
Eucalyptol ^Me^	14.046	0.64
γ-terpinene ^Mh^	14.872	0.11
Terpinolene ^Mh^	15.740	0.09
α-Thujone ^Mk^	15.898	0.11
Menth-2-ene-1-ol-Cis para ^Ma^	17.010	0.81
Menth-2-ene-1-ol-Trans para ^Ma^	17.556	0.55
P-Cymene-8-ol ^Ma^	18.897	0.12
α-terpineol ^Ma^	19.121	1.06
Cis Piperitol ^Ma^	19.517	0.28
Piperitone ^Mk^	20.915	49.90
β-bourbonene ^Sh^	24.192	0.11
β-elemene ^Sh^	24.313	1.50
β-caryophyllene ^Sh^	25.104	1.26
β-gurjunene ^Sh^	25.371	0.21
Aromadendrene ^Sh^	25.561	0.20
α-humulene ^Sh^	25.986	0.14
Allo-aromadandrene ^Sh^	26.090	0.12
Germacrene-D ^Sh^	26.598	0.28
β-Selinene ^Sh^	26.804	0.32
α-selinene ^Sh^	26.955	0.34
Cuparene ^Sh^	27.262	0.22
γ-cadinene ^Sh^	27.344	0.12
δ-cadinene ^Sh^	27.443	0.35
Elemol ^Sa^	28.183	5.79
Spathulenol ^Sa^	28.844	0.15
Caryophyllene oxide ^So^	28.989	0.33
Viridiflorol ^Sa^	29.059	0.12
γ-Eudesmol ^Sa^	29.889	0.10
Epi- γ-eudesmol 10 ^Sa^	30.072	0.83
α-Eudesmol ^Sa^	30.607	2.65
Total		98.46
Monoterpene hydrocarbons		29.73
Monoterpene ethers		0.75
Monoterpene ketones		50.01
Monoterpene alcohols		2.82
Sesquiterpene hydrocarbons		5.18
Sesquiterpene alcohols		9.64
Sesquiterpene oxides		0.33

The initial of each molecule corresponds to the chemical group to which it belongs in the lower part of the table. ^Mh^, Monoterpene hydrocarbons; ^Me^, Monoterpene ethers; ^Mk^, Monoterpene ketones; ^Ma^, Monoterpene alcohols; ^Sh^, Sesquiterpene hydrocarbons; ^Sa^, Sesquiterpene alcohols; ^So^, Sesquiterpene oxides.

**Table 2 molecules-28-04561-t002:** Antioxidant activity of *C. schoenanthus* EO.

Compound Tested	DPPH Test	ABTS Test
EO of *C. schoenanthus*	1730.00 ± 80.00	2890.00 ± 26.90
Trolox	1.84 ± 0.07 ***	7.57 ± 1.8 ***

DPPH, (2,2-diphenyl-1-picrylhydrazyl); ABTS, (2,2′-azino-bis-[3-ethylbenzothiazoline-6-sulfonic acid]); values are expressed as mean values ± SD. *n* = 3 independent experiments in quadruplicate for the measurement of antioxidant activity; DPPH and ABTS activities are expressed as IC_50_ (µg/mL); *** *p* < 0.05, values significantly different for each test compared to the positive control Trolox; EO, essential oil.

**Table 3 molecules-28-04561-t003:** IC_50_ (µg/mL) of EO of *Cymbopogon schoenanthus* tested on LNCaP human prostate cancer and HeLa human cervical cancer cell lines.

Compound Tested	LNCaP Cells	HeLa Cells
EO of *C. schoenanthus*	135.53 ± 5.27	146.17 ± 11.83
Cisplatin	3.20 ± 0.50 ***	5.20 ± 0.80 ***

Values are expressed as mean ± standard deviation; *n* = 3 independent experiments in sextuplicate; EO, essential oil; *** (*p* < 0.001), values significantly different compared to chemotherapeutic agent cisplatin (positive control) for each cell line.

## Data Availability

Datasets generated during the current study are available from the corresponding author upon reasonable request.

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
