# Peer review of "Chemical Composition of Essential Oil of Cymbopogon schoenanthus (L.) Spreng from Burkina Faso, and Effects against Prostate and Cervical Cancer Cell Lines"

_molecules, 2023, doi:10.3390/molecules28114561_

Round 1

Reviewer 1 Report

The study entitled „Chemical composition of essential oil of Cymbopogon schoenanthus (L.) Spreng from Burkina Faso, and effects against prostate and cervical cancer cell lines“ is a useful original study. The article fulfils all the formal requirements for publication in Molecules. However, before publication, several comments must be implemented.

Lines 137-138 in the discussion; mention the type of used cancer cells,

Lines 163-173, please discuss your data with other authors,

The topic of the article is plant extract in cancer management; therefore, Conclusions must be improved in this regard! I suggest the authors implement the concept of traditional medicine in cancer management where the anticancer effects of combining various natural compounds to target a range of signalling pathways within cancer cells should be superior compared to a single compound targeting only one signalling pathway. While low doses of a single compound may not achieve the full anti-cancer action, a mixture of compounds, or extracts from medical plants, may be able to specifically synergize in the prevention/therapy of cancer with negligible side effects. This clinical approach could be specifically used for indications such as cancer prevention in high-risk individuals or in the combined anti-cancer therapy to re-sensitize cancer.

Within this concept, the important citation should be used: Int J Mol Sci. 2020 Dec 26;22(1):183. doi: 10.3390/ijms22010183. PMID: 33375383; PMCID: PMC7795985.

The text is easy to understand, written by experienced scientists.

Author Response

The study entitled „Chemical composition of essential oil of Cymbopogon schoenanthus (L.) Spreng from Burkina Faso, and effects against prostate and cervical cancer cell lines“ is a useful original study. ** We would like to thank Reviewer#1 for this positive comment. The modifications have been enlighten in yellow.  

Lines 137-138 in the discussion; mention the type of used cancer cells

** As suggested, the cell lines used by Mohamed Abdoul-Latif et al. have been mentioned please note the new numbering of the lines.

Lines 163-173, please discuss your data with other authors

** As suggested, data on migration and cell cycle of LNCa cells have discussed by citing new references 19 and 20. We hope this will be OK with Reviewer#1. 

I suggest the authors implement the concept of traditional medicine in cancer management where the anticancer effects of combining various natural compounds to target a range of signalling pathways within cancer cells should be superior compared to a single compound targeting only one signalling pathway. 

** we would like to thank Reviewer#1 for this helpful suggestion. As recommended, we have inserted this point in the Conclusion part. Moreover, Kubatka's paper has been added in the list of references. 

Reviewer 2 Report

Please find comments in the attached file.

Author Response

 The paper is written well and may be a good reference for essential oil chemical investigation, antioxidant, and cytotoxicity studies. 

** We would like to thank Reviewer#2 for these positive comments. The modifications based on the comments have been enlightened in green in the revised manuscript.

 The introduction: Please specify what was the logic for examining your plant extract against these cell lines specifically. 

** As suggested we have indicated that prostate and cervix cancers were among the most frequent  in Burkina Faso (GLOBOCAN) (lines 64-68 and 70-71).

Overall, the plant name should be italicized through the manuscript. ** As suggested by Reviewer#2, plant names have been italicized through out the manuscript.   In Table 1, the lower part is not easily confirmed from just names of the molecules in the upper part of the table; I recommend adding the compound structure as supplementary file and refer to that in the footnote of the table. ** We agree that the lower part was not that simple to identify the compound just form the names. We have thus indicated in the Table the various families: Mh, Me, Mk, Ma, Sh, Sa, So. This seemed easier for the readers than adding a supplementary data with the structures. We hope that this solution will satisfy Reviewer#2.   Cisplatine in line 128 is not correct. ** The name was corrected accordingly (line 145).   Section 3.5. and 3.6. Measurement of cell survival should be used once, and the different experimental followed may be represented as subheading “a and b”. ** We apologize for this mistake. Section 3.6 was corrected accordingly and named as indicated (one 281).   The conclusion section should be modified. Your cytotoxicity results are not significant. ** Actually, we disagree with this statemen, results are significant but less than cisplatin. This was already indicated in lines 165 and 166. However, we have taken into account Reviewer#2's concern and have even more minimized the effects of the EO (lines 144).   The conclusion “Altogether, this work constitutes a scientific basis for the use of EO of C. schoenanthus from Burkina Faso for the traditional management of tumors.” Is too much bolder than the observed findings; please justify your conclusion. ** As also reported by Reviewers #1 and #3, we have modified the last paragraph of the manuscript (lines 309-331) in order to minimize our conclusions. We hope the modifications will be in accordance with Reviewer#2's comments.  

Reviewer 3 Report

A positive control should be included in the flow cytometry analysis.

In its present form, this manuscript seems preliminary. The authors are encouraged to perform cytotoxic experiments with piperitone, the main compound of this essential oil. In addition, additional experiments (e.g., western blot analysis, immunohistochemistry, etc.) are necessary to evaluate a possible mechanism of the cytotoxic activity of this compound. Probably, the authors are considering a second manuscript with this compound. I recommend including this information in the present manuscript.

What are the uses of this plant species in traditional medicine? This information should be included in the introduction section.

With the information here presented, I do not consider the authors can conclude that the "EO of C. schoenanthus from Burkina Faso for the traditional management of tumors". For this conclusion, the authors should perform an in vivo study.

No comments

Author Response

The authors are encouraged to perform cytotoxic experiments with piperitone, the main compound of this essential oil. In addition, additional experiments (e.g., western blot analysis, immunohistochemistry, etc.) are necessary to evaluate a possible mechanism of the cytotoxic activity of this compound. Probably, the authors are considering a second manuscript with this compound. I recommend including this information in the present manuscript.

** As indicated in the submission letter, this work is a joint project between France and Burkina Faso. Unfortunately due to political turmoil in Faso, it is difficult to plan new experiments in the next few months.  We understand Reviewer#3 reluctance according our strong delusions about the use of the studied EO to treat cancer. We have thus tried to amend the manuscript, being less affirmative regarding the translational use of the compound and on the contrary, being more critics regarding the effects ex vivo. We thus have drastically modify the conclusion 

What are the uses of this plant species in traditional medicine? This information should be included in the introduction section.

** As requested by Reviewer#3, this information was introduced, highlighted in blue, as well as two new references.

With the information here presented, I do not consider the authors can conclude that the "EO of C. schoenanthus from Burkina Faso for the traditional management of tumors". For this conclusion, the authors should perform an in vivo study.

** As also recommended by Reviewer#1, and according Reviewer#3's concern, we have amended the conclusion (lines 321-331), indicating that this work is a starting point to go further in order to characterize in vivo the effects of C. schoenanthus EO.

A positive control should be included in the flow cytometry analysis.

** Unfortunately as in our previous article on that topic we only added a negative control for the experiments (DMSO). This has never pointed out before. We however agree with Reviewer#3 with this concern. In the future we will apply that recommandation. 

Round 2

Reviewer 3 Report

Dear authors

I understand that the political issues in your country are not up to you. However, without additional experiments, the manuscript seems preliminary for a journal like Molecules.

A positive control is necessary for the flow cytometry analysis.

No comments

Author Response

As discussed in our previous response, we can only agree with Reviewer 3 about the fact that more experiments should be performed. However, we should also state that we have not performed less experiments than other groups on the same kinds of research article published in hight impact journals such as Molecules.   We would like to remind that the core of our paper is that EO from C. schoenanthus has clear effects on cell lines derived from cervix and prostate tumors. We have never tried to overly sell the data, and this the reason why we have also minimized our conclusions compared to the first submitted manuscript.    * Indeed, it could be a fantastic idea to perform in vivo experiments using preclinical models as suggested by Reviewer #3. However, this will not be done, and this for mainly two reasons: 1) France, and globally Europe, is engaged on the 3R process to abolish unnecessary experiments on animals. Doing the proposed experiments would need to fulfill an ethic protocol that is long to obtain and at least 6 months to obtain the animals, then perform the experiments and analyze the data. 2) the funds necessary for such experiments have nothing to do with what we have already done for this work, specially regarding the technical aspects.   * In all our previous works published, no reviewer has ever asked a "positive" control for the cell cycle analysis. This is not surprising: we have showed the effect of a compound (here EO from C. schoenanthus) vs. the "placebo" condition, i.e., the vehicle (here DMSO). We could eventually perform new experiments on LNCaP cells exposed to Cisplatin, but the result would indicate that cisplatin has an effect on LNCaP cells, and so what... Because EO significantly modifies the cell cycle, we do know that we are able to visualize any changes in the cell cycle.    * Performing western blot and qPCR, and deciphering the apoptotic mechanism would also be a great objective. However, doing this is a totally new project that we cannot go in for funding reasons.   * Testing the effect of piperitone seems to be out of the bulk of this article. Only one article reports the effect of this compound on DU145 prostate cells, with an IC50 of 30.51 ± 0.18 µg/ml.   However, and despite all the points developed above, we want to prove our good faith and are ready to perform the cytotoxic assays with piperitone. We have ordered some and MTS assays will be done ASAP. Obviously, receiving the product, performing 3 independent experiments, will need more than 10 days.   Because we do respect Reviewer#3's comments and since we do not want to bargain the decision as Molecules has its own editorial policy and our groups have also their own obligations, please feel free to accept or reject this proposition. In the second option, we will send this article to another journal.